# Systematic Review with Meta-Analysis: Comparison of the Risk of Hepatocellular Carcinoma in Antiviral-Naive Chronic Hepatitis B Patients Treated with Entecavir versus Tenofovir: The Devil in the Detail

**DOI:** 10.3390/cancers14112617

**Published:** 2022-05-25

**Authors:** Hyunwoo Oh, Hyo Young Lee, Jihye Kim, Yoon Jun Kim

**Affiliations:** 1Department of Internal Medicine, Uijeongbu Eulji Medical Center, Eulji University School of Medicine, Uijeongbu 11759, Korea; asklepios1258@eulji.ac.kr (H.O.); 2hyo0@eulji.ac.kr (H.Y.L.); 2Department of Internal Medicine, Seoul National University Bundang Hospital, Seongnam 13620, Korea; stjihye@snubh.org; 3Department of Internal Medicine, Liver Research Institute, Seoul National University College of Medicine, Seoul 03080, Korea

**Keywords:** tenofovir, entecavir, hepatocellular carcinoma, antiviral-naïve, meta-analysis

## Abstract

**Simple Summary:**

Tenofovir disoproxil fumarate (TDF) and entecavir (ETV) are the preferred anti-viral agents used as first-line treatments for chronic hepatitis B. Despite many meta-analyses being conducted, it is still not clear whether TDF is more effective than ETV at reducing the risk of HCC due to the inconsistent statistical methodologies employed in previous observational studies. To reduce heterogeneity, we analysed only hospital cohort data studies with anti-viral naive patients. Additionally, unlike previous studies, we conducted subgroup analyses with enrolment criteria and socioeconomic factors that could not be corrected with statistical techniques. There is no difference between the two drugs in terms of reducing the risk of HCC in a pooled analysis of PS-matched patients. In the subgroup analysis, if there was interval of over three years from the start point of patient enrolment, we found that TDF was associated with significantly lower HCC risk. This result will provide new perspectives for future research.

**Abstract:**

Tenofovir disoproxil fumarate (TDF) and entecavir (ETV) are the preferred anti-viral agents used as first-line treatments for chronic hepatitis B (CHB). However, the efficacy of these agents in reducing the incidence of hepatocellular carcinoma (HCC) remains unclear. We conducted this meta-analysis to assess the efficacy of anti-viral agent on preventing HCC in CHB. Two investigators independently searched all relevant studies that examined the efficacy of anti-viral agent for preventing HCC using MEDLINE, Embase, and Cochrane Library databases through August 2021. The extracted data were analysed using a random-effects meta-analysis model based on the inverse-variance method (DerSimonian–Laird) and expressed as hazard ratio (HR) and 95% confidence interval (95% CI). We included 19 retrospective studies in the analysis. Although there was substantial heterogeneity between the studies, the overall pooled HR indicated that TDF significantly lowered the risk of HCC (HR: 0.72, 95% CI: 0.58–0.90, I^2^ = 66.29%). However, the pooled analysis of propensity score (PS)-matched subpopulations showed no significant differences (HR, 0.83; 95% CI, 0.65–1.06; I^2^ = 52.30%) between TDF and ETV. In a subgroup analysis, an interval of over three years in the start point of patient enrolment and excluding alcoholic liver disease patients significantly lowered the HCC risk associated with TDF. In conclusion, TDF may be more effective than ETV at reducing HCC incidence in treatment-naive CHB patients, but this effect was not consistent in the PS-matched subpopulation that reduced heterogeneity. As a result of subgroup analysis, the conflicting findings of previous studies may result from heterogeneous inclusion criteria. Further studies with standardised protocols are needed to reduce the residual heterogeneity.

## 1. Introduction

Chronic hepatitis B (CHB) infection is one of the most common causes of chronic liver disease, affecting approximately 300 million patients worldwide. According to the World Health organisation, CHB caused an estimated 820,000 deaths from cirrhosis and hepatocellular carcinoma (HCC) in 2019 [1]. With the development of hepatitis B antiviral agents and the inhibition of hepatitis B virus (HBV) replication with long-term nucleos(t)ide analogue (NA) therapy, the overall survival of CHB patients has increased [2]. However, the risk of HCC persists [3]. In a real-world clinic, a lifetime prescription of medication for CHB is a critical issue and should be based on a high level of evidence. However, prescribing drugs that reduce the risk of HCC can contribute to reducing socioeconomic costs [4].

Among the available NA therapies, entecavir (ETV) and tenofovir disoproxil fumarate (TDF) are both recommended as first-line treatments for CHB [5,6]. Many conflicting studies have been published since Choi et al. reported a low risk of HCC in a TDF user group within a CHB hospital cohort and South Korea’s nationwide claim data [7]. However, previous studies have been highly heterogeneous in terms of baseline characteristics, follow-up duration, use of other NAs, and statistical methodology, making it difficult to make objective comparisons. Similarly, previous meta-analyses [8,9,10,11,12,13,14,15,16,17,18,19,20] have also failed to reach an agreement owing to the following limitations: pooled analysis of odds ratios with different follow-up durations [21], a mix of antiviral naïve and non-naïve patients, and pooled analysis of hazard ratios (HRs) using retrospective hospital cohort data and administrative databases or claim data at once. These inconsistent statistical methodologies of the previous studies were pointed out in a recent review article [22].

Therefore, whether TDF is more effective than ETV at reducing the risk of HCC remains inconclusive. This systematic review and meta-analysis aims to compensate for the limitations of previous studies and obtain new insights into the efficacies of TDF and ETV on incidence of HCC in CHB patients.

## 2. Materials and Methods

### 2.1. Data and Literature Source

Two investigators (Hyunwoo Oh and Hyo Young Lee, Department of Internal Medicine, Eulji University School of Medicine, Uijeongbu, Korea) independently searched the MEDLINE, EMBASE, and Cochrane Library databases using the following keywords: “tenofovir”, “entecavir”, and “hepatocellular carcinoma”. Additional references were obtained from the bibliographies of relevant articles published through 31 August 2021 (Table 1). There was 96.4% agreement between the reviewers regarding the eligibility of articles after full-text screening, corresponding to a substantial agreement (k = 0.867). Any disagreement or unresolved concern was independently reviewed by the corresponding author (Figure 1).

### 2.2. Inclusion and Exclusion Criteria

The inclusion criteria for the study selection were (1) antiviral-naive patients with CHB over 18 years of age; (2) human subject study design including randomised control trials (RCTs) and non-RCTs with two arms of either ETV or TDF monotherapy; and (3) suggesting the risk of HCC development with HR as a primary or secondary outcome.

Studies on (1) co-infection with other hepatotropic viruses (i.e., hepatitis C, D, or E virus) or human immunodeficiency virus; (2) unreported HCC incidence in either the TDF or ETV arm; (3) combination antiviral therapy or sequential therapy; and (4) observational retrospective cohort studies using administrative database or medical claim data were excluded from our analysis.

### 2.3. Data Extraction

Two investigators independently extracted data from each study using a predefined electronic spreadsheet to minimise random and bias errors. Any disagreement or unresolved concerns were independently reviewed by the corresponding author. If necessary, we contacted the co-author or corresponding author to rule out uncertainty (no mention of reference value being selected for multivariable Cox proportional hazards model: ETV or TDF [30,31]). As a result, all HRs were presented for the excess risk of each outcome among patients treated with TDF compared to ETV (extracted and calculated using ETV as a reference value).

### 2.4. Assessment of Methodological Quality

Two investigators independently evaluated the quality of the included studies using the Newcastle–Ottawa scale (NOS) for non-randomised studies (Appendix A) [41]. Any disagreement or unresolved concerns were independently reviewed by the corresponding author.

### 2.5. Statistical Analysis

Extracted data were analysed with the inverse variance (IV) using the natural log of HRs as described by Parmar et al. [42] and the DerSimonian–Laird random-effects model for the meta-analysis. Heterogeneity across the enrolled studies was investigated using the Cochran Q test and Higgins I^2^ value. I^2^ values exceeding 25%, 50%, and 75% represent low, moderate, and high heterogeneity, respectively. The level of significance for the test for heterogeneity was investigated using the chi-squared test [43]. Potential sources of heterogeneity were investigated using subgroup analyses with commonly applied enrolment criteria in the included studies and the start point of patient enrolment not reflected in enrolment criteria. To evaluate the source of heterogeneity, we examined the adopted variables for univariate and multivariate Cox regression analysis and propensity score matching (PSM) analysis (Appendix A). We also collected and compared the statistical techniques used in the studies, including methods for variable selection for Cox regression analysis, *p*-value cut-off for variable selection in the multivariate model, PSM method and calliper size, inverse probability treatment weighting (IPTW), competing for risk analysis, and multiple imputations for missing data. We used a funnel plot to visualise the publication bias. Using the arcsine Thompson’s (AS-Thompson’s) test, we evaluated the funnel plot asymmetry due to the high heterogeneity of the enrolled studies [44]. Statistical analyses were performed using R statistical software (version 3.6.3 (accessed on 29 February 2020)); R Foundation, Inc, Vienna, Austria.; (http://cran.r-project.org (accessed on 24 May 2022)) R package ‘meta’ and ‘metasens’.

The present systematic review of the literature was performed based on the Preferred Reporting Items for Systematic Reviews and Meta-Analysis (PRISMA) statement and checklist [45]. Patient consent and Institutional Review Board approval were not required because this was a systematic review of already published articles. This study is registered with the Open Science Framework (https://osf.io/ (accessed on 24 May 2022)), and its unique identifying number is: 10.17605/OSF.IO/964UA.

## 3. Results

Nineteen out of 1733 studies were included in the final meta-analysis. All included studies were observational retrospective cohort studies with 57,455 antiviral-naïve patients from hospital cohorts (Figure 1, Appendix A). All studies were reported between 2017 and 2021, and 12 out of 19 studies were conducted in Korea (Table 1). The number of enrolled patients in major countries was 30,858 in Hong Kong, 18,684 in South Korea, 5565 in Taiwan, and 1819 in the U.S.A. The different studies had diverse inclusion and exclusion criteria (Table 2). The TDF and ETV treatment groups in these studies differed in terms of the time of treatment initiation (calendar year) and risk factors (host, hepatic, and viral). The studies used different variables for univariate and multivariable Cox regression analyses of the risk of HCC development and PSM analysis (Appendix A). In addition, the statistical methods used in the included articles were diverse (Appendix A). All studies scored six to eight stars in the NOS, indicating satisfactory quality (Appendix A).

### 3.1. Pooled Analysis of Representative HRs Presented in Individual Papers

The pooled HR of 19 studies for HCC development with TDF over ETV monotherapy was 0.72 (95% confidence interval [CI], 0.58–0.90, *p* < 0.01) (Figure 2), indicating a significantly lower HR for HCC development in the TDF group than in the ETV group. However, the outcomes of the included studies showed substantial heterogeneity (I^2^ = 66.29%, *p* < 0.01) (Figure 2). The AS-Thompson test for publication bias found no significant asymmetry in the funnel plot (*p* > 0.1) (Appendix A).

### 3.2. Adjusted HR by Multivariable Analysis

Compared to the pooled HR of representative HRs from studies that presented adjusted HR, the adjusted pooled HR of 11 studies was 0.75 (TDF vs. ETV; 95% CI, 0.64–0.88; *p* < 0.01). This result indicates that the HR for HCC development in the TDF group was significantly lower than that in the ETV group. No significant heterogeneity was detected using the Q-test (I^2^ = 24%, *p* = 0.21) (Figure 3). The AS-Thompson test for publication bias found no significant asymmetry in the funnel plot (*p* > 0.1) (Appendix A). None of the variables were commonly adopted for multivariable analysis in all included studies (Appendix A).

### 3.3. PS-Matched Population

In contrast to the pooled analysis of representative HRs and adjusted HRs, pooled analysis of the PS-matched sub-cohort with 10 studies showed no significant difference between the two groups (from HR: 0.65, 95% CI: 0.47–0.90, *p* < 0.05 to HR: 0.83, 95% CI: 0.65–1.06, *p* = 0.13). Substantial heterogeneity (I^2^ = 52%, *p* = 0.03) was detected in the outcomes (Figure 4). The number of subjects decreased from 49,706 to 20,151 after PSM (Appendix A). No significant asymmetry in the funnel plot was observed using the AS-Thompson test for publication bias (*p* > 0.1) (Appendix A).

### 3.4. Cirrhotic Subcohort

In CHB cirrhosis sub-cohorts, the pooled representative HRs showed a significantly lower risk of HCC development in the TDF group than in the ETV group (HR: 0.75, 95% CI: 0.58–0.96, *p* = 0.02). However, this was not consistent with the findings of the pooled analysis with adjusted HR (HR: 0.80, 95% CI: 0.64–1.00, *p* = 0.054) or the PS-matched population (HR: 0.95, 95% CI: 0.78–1.16, *p* = 0.632). Since fewer than 10 studies were included, the results should be interpreted with caution (Appendix A).

### 3.5. Subgroup Analysis

To determine the cause of heterogeneity, a subgroup analysis was conducted based on each study design and patient enrolment criterion. An interval of over three years in the start points of patient enrolment (or U.S. Food and Drug Administration (FDA) approval date of TDF and ETV) between the two groups resulted in a lower risk of HCC development in the TDF group than that in the ETV group (HR: 0.83, 95% CI: 0.62–1.12 vs. HR: 0.69, 95% CI: 0.51–0.92) (Table 3, Appendix A). Additionally, the exclusion of patients with significant alcoholic liver disease lowered the risk of developing HCC in the TDF group compared to the ETV group (HR: 0.58, 95% CI: 0.44–0.76, *p* < 0.01) (Table 3, Appendix A). All four studies that excluded patients with significant alcoholic liver disease were conducted in Taiwan.

## 4. Discussion

The novelty of this study lies in the fact that we extracted and analysed data from only antiviral-naïve CHB patients. In a 12-year follow-up cohort study, Papatheodoridis et al. found a significant difference in the development of HCC in NA-naïve (67/1128; 5.9%) vs. NA-experienced (76/807 or 9.4%) patients (*p* = 0.004) [46]. Since we analysed only NA-naïve CHB patients, there was no concern about ETV resistance caused by previous drug exposure, thereby reducing the heterogeneity when comparing the effects of drugs.

In our study, the significance of pooled HR was negligible in the PS-matched population when compared with the representative HRs presented in individual papers with adjusted HRs. Therefore, it is important to determine the compounding factors that reduce heterogeneity in adjusted, PS-matched subpopulations and affect HCC development other than drug choice.

In the subgroup analysis, we observed significant differences in the clinical outcomes of the two groups due to differences in patient enrolment timing (Table 3). After the FDA approval of the two drugs (ETV 2005 and TDF 2008), there have been many modifications to the international treatment guidelines, and the indications for the application of NAs vary from country to country (Figure 5). Although the analysis methods are different, inconsistency in clinical outcomes could arise from disparities in the follow-up length, as discussed in a similar meta-analysis [8]. Additionally, a study by Chen et al. addresses the implications of this disparity [39]. Taiwan is a country with a National Health Insurance system, and TDF has been included in the benefits eligibility since 2011 (Appendix A) [47]. Chen et al. found that TDF treatment was associated with a lower risk of HCC in the entire (*n* = 1560, HR: 0.585, 95% CI: 0.425–0.806, *p* < 0.001) and treatment-naïve (*n* = 1353, HR: 0.523, 95% CI: 0.363–0.752, *p* < 0.005) cohorts [39]. However, a subgroup analysis of patients (not restricted to naive patients only) enrolled after 2011 did not find a lower risk of HCC (*n* = 1162, HR: 1.987 95% CI: 1.392–2.837, *p* < 0.001). Before reimbursements for TDF treatments began, CHB patients with a relatively high risk of developing HCC and waiting to be reimbursed for antiviral treatments started to take ETV, which may account for the higher incidence of HCC in the ETV group.

Similar to Taiwan, South Korea also has a National Health Insurance system. Twelve studies in this analysis (63%) included CHB patients from South Korea. In South Korea, when the TDF reimbursement benefits were available, the indications for its use were eased compared to those for ETV. As a result, the severity of the antiviral treated patient group decreased. A cohort study comparing the ETV and TDF groups by year of enrolment or reimbursement policy should be designed to demonstrate this. Oh et al. designed a study to reduce the influence of reimbursement policy, controlling the treatment start date so that the same criteria were applied. They found that TDF treatment was not associated with a lower risk of HCC (HR: 1.26, 95% CI: 0.81–1.97, *p* = 0.303). However, a limitation of their study was that the treatment starting date for the two groups did not match [36].

A subgroup analysis also showed a significantly lower risk of HCC development in the TDF group when patients with alcoholic liver disease were excluded from the study. All of these studies were conducted in Taiwan. Although not significant, Taiwanese studies included a relatively higher proportion of male CHB patients (>70%) than studies from other countries. If there is a difference even in patients with alcoholic liver disease with a relatively high risk of HCC due to high drinking and smoking rates, the characteristics of the female group may have contributed to it, given the preference for TDF in women of childbearing age, which may have caused a bias [48].

There was substantial heterogeneity in the enrolled studies. One of the causes for this is the differences in their inclusion and exclusion criteria. First, while some papers excluded CHB patients who developed HCC within six months, others excluded those who developed HCC within one year. It has been reported that the tumour volume doubling time (TVDT) of HCC is approximately 4–5 months [49], and regular HCC surveillance is typically implemented every six months. It is therefore difficult to rule out the possibility that HCC present at the start of the treatment was not included by excluding patients who developed HCC within six months. Second, patients with baseline HBV DNA levels of <2000 IU/mL were excluded in several studies [7,24,28,35]. Four such studies report contradictory statistical significance, possibly due to differences in their exclusion criteria. In Korea, which accounted for 63% of the studies in the present meta-analysis, if patients with baseline HBV DNA levels of <2000 IU/mL were excluded, decompensated cirrhosis patients would have been automatically excluded owing to the change in reimbursement policy since September 2015 (Appendix A), resulting in a significant difference in HCC risk between the TDF and ETV groups.

Moreover, comorbidities in CHB patients could affect the development of HCC. Patients with chronic kidney disease (CKD) and osteoporosis might have been prioritised for ETV treatment due to safety issues, even when both drugs were available for prescription [6]. In the articles included in this study, before PS matching, the age and comorbidities, including hypertension, diabetes, and CKD, were higher in the ETV group, although not significant. In a recent case-control study using medical claims data from South Korea, the proportion of patients with CKD was higher among those with CHB than among matched controls (3.02% vs. 1.14%, *p* < 0.01) [50]. In a retrospective observational study, patients with stages 4 and 5 CKD showed a higher incidence of HCC, although the cohort included patients with chronic hepatitis C and hepatitis B and C co-infected patients [51].

Medication compliance was also an important covariate. Very few studies in the present meta-analysis adopted the cumulative defined daily dose (cDDD) as a covariate [25,37]. Medication compliance should be treated as an important compounding factor, as poor compliance affects the emergence of the ETV mutant, which is one of the risk factors for HCC development [52]. Unlike TDF, ETV is a pre-meal drug, and its pre-prandial administration is associated with non-adherence [53]. In the enrolled studies, the ETV group appeared to have a relatively low cDDD [25,37]. In a study by Choi et al. [7], the treatment modification rate was significantly higher among ETV users than among TDF users (182/1560; 11.7% vs. 2/1141; 0.2%). In South Korea, replacing or switching to other NAs is difficult because of reimbursement issues unless physicians prove drug resistance, insufficient treatment response, pregnancy, or serious side effects through documentation. Therefore, for the reasons listed above, it can be assumed that the treatment response in the ETV user group was poor.

Differences in methods of statistical analysis were also considered. Previous studies have used several techniques, such as PSM, IPTW, multiple imputations, and competing risk analysis. The Cox proportional hazards model can adopt a direction/forward/backward stepwise method. Depending on the study design, both direction methods are generally recommended; however, most enrolled studies did not mention which method was chosen (Appendix A).

A limitation of retrospective cohort studies is that several unobservable confounding factors may be present. Even if residual heterogeneity is allowed, patients may experience deterioration of cirrhosis owing to lifestyle [54], alcohol consumption, or poor medication compliance. Moreover, it is well known that fatty liver disease [55], family history [56], concomitant medications [57,58], and exposure to aflatoxin B1 [59] affect HCC incidence. These above factors, causing inflammatory reactions, can stimulate the multistep process of hepatocarcinogenesis [60]. Therefore, predicting HCC development based on the baseline characteristics without considering events occurring during the observation period can lead to inaccuracy. In a recent study, Lee et al. suggested that the presence of cirrhosis at the time of HBeAg seroclearance could be a compounding factor for HCC development [61].

This study has considerable limitations. About 97% of patients were derived from Asia, and thus, the characteristics of patients in this study may be different from Caucasian cohorts. In a recent study comparing 9143 Korean and 719 Caucasian CHB patients, Jang et al. found higher HBeAg positivity among Koreans (49.1%) than Caucasians (20.3%). Nevertheless, the HBeAg-positive phase occurs early in the natural course, while the proportion of LC was also higher among Koreans (41.1%) than Caucasians (31.5%) [62]. Moreover, the majority of studies did not present virologic data, such as the genotype of HBV, which is known to have different geographical distributions [63]. Therefore, HBeAg status as well as various characteristics such as cirrhosis status and genotype of HBV should be taken into account when interpreting the results.

## 5. Conclusions

In conclusion, our analysis found that the incidence of HCC following TDF monotherapy was significantly lower than after ETV monotherapy with high heterogeneity. However, this difference was not seen with a pooled HR in a PS-matched sub-cohort that reduced the heterogeneity of the TDF and ETV user groups. There are many observable and unobservable confounding factors that can affect the heterogeneity of these studies. Even with several statistical techniques, such as PSM analysis, socioeconomic factors such as reimbursement policies may not be corrected. As a limitation of retrospective-cohort studies, there is not enough data to establish different efficacies of TDF and ETV on incidence of HCC in CHB patients. Therefore, further prospective studies with standardised protocols or individual patient data meta-analyses are needed to reduce the residual heterogeneity that may affect HCC development by mechanisms other than drug choice.

## Figures and Tables

**Figure 1 cancers-14-02617-f001:**
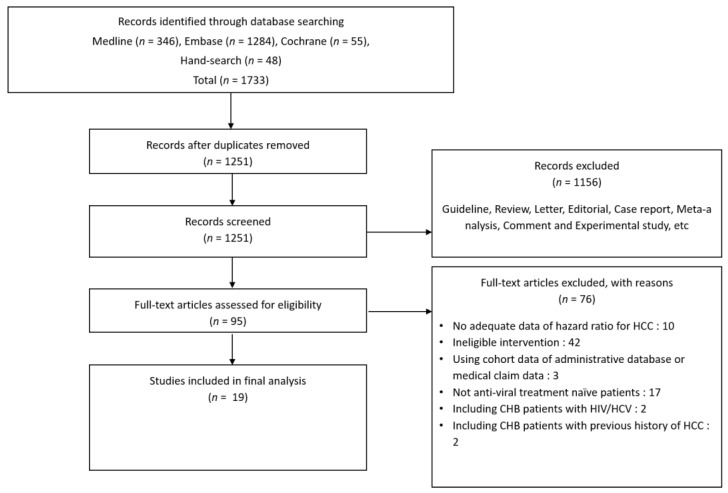
Flow diagram showing the literature search (31 August 2021 record).

**Figure 2 cancers-14-02617-f002:**
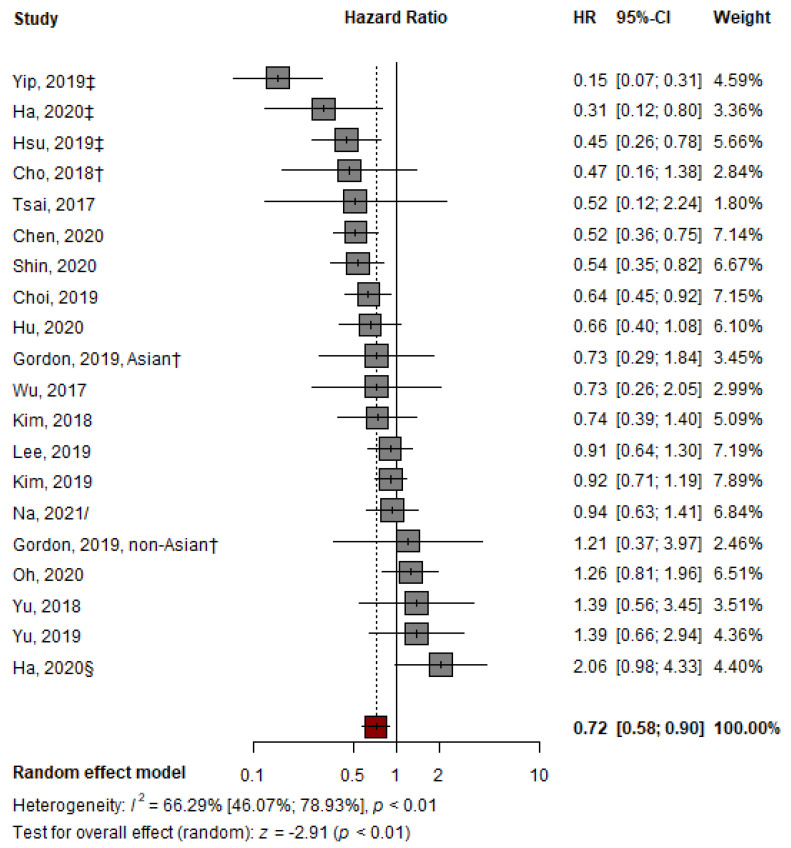
Pooled analysis of representative HRs presented in individual papers comparing the effectiveness of TDF vs. ETV at reducing HCC development. HR: hazard ratio; † abstract; ‡ suggest outcomes from competing risk analysis; § Ha from CHA Bundang Medical Center, CHA University; / from unadjusted cohort at the time of CVR [7,23,24,25,26,27,28,29,30,31,32,33,34,35,36,37,38,39,40].

**Figure 3 cancers-14-02617-f003:**
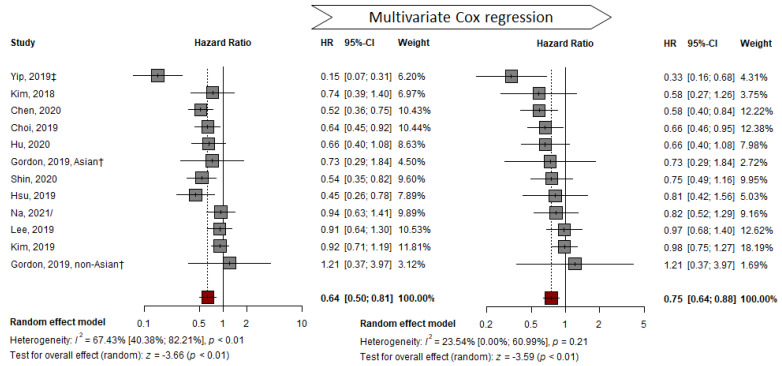
Multivariable adjusted HR pooled analysis comparing the effectiveness of TDF vs. ETV at reducing HCC development. HR: hazard ratio; † abstract; ‡ suggest outcomes from competing risk analysis; / from unadjusted cohort at the time of CVR [7,23,24,25,26,27,29,33,38,39,40].

**Figure 4 cancers-14-02617-f004:**
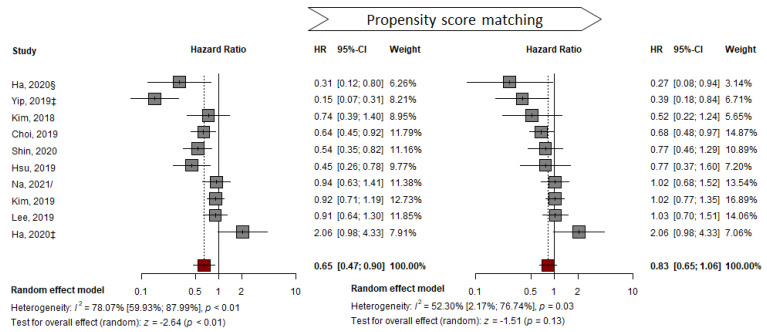
Propensity-score matched HR pooled analysis comparing the effectiveness of TDF vs. ETV at reducing HCC development. ‡ suggest outcomes from competing risk analysis; § Ha from CHA Bundang Medical Center, CHA University; / from unadjusted cohort at the time of CVR [7,24,25,26,27,29,33,35,37,40].

**Figure 5 cancers-14-02617-f005:**
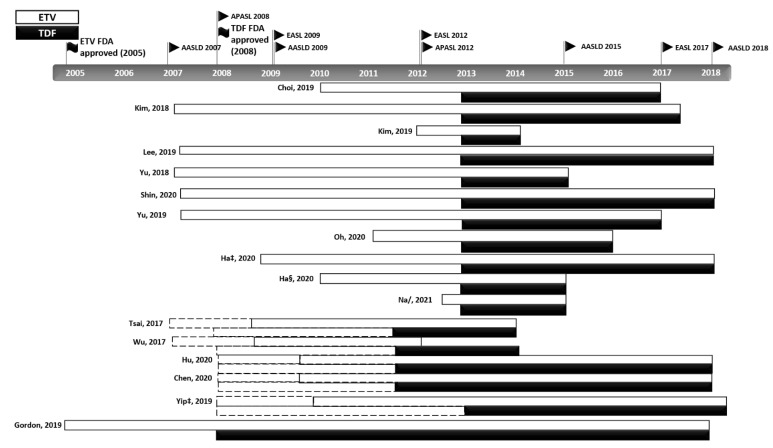
Modifications to the international treatment guidelines for CHB. The various changes and updates to the CHB guidelines and reimbursement policies and differences in the start dates of patient enrolment are shown. Dotted line: drug prescription available; solid line: drug prescription with reimbursement benefits available. ‡ suggest outcomes from competing risk analysis; § Ha from CHA Bundang Medical Center, CHA University; / from unadjusted cohort at the time of CVR [7,23,24,25,26,27,28,29,30,31,32,35,36,37,38,39,40].

**Table 1 cancers-14-02617-t001:** Main characteristics of the included studies (all studies were retrospective observational studies, and the number of decimal places is borrowed from the adopted articles).

Author Year Country	Enrolment Year	Duration (Months) TDF ETV	Cirrhosis (%)	Patients (*n*)	Age (Mean (±SD))	Sex (Male%)	HBV_DNA	HBeAg Positive	HR (Naïve Only)
				TDF ETV	TDF ETV	TDF ETV	(log10) (IU/mL)	TDF (%) ETV (%)	(Ref: ETV)
Choi 2019 Korea [7]	2010.01–2016.12	32 (23–43) 48 (36–48)	653 (57.2) 935 (59.9)	1560 1141	48.1 ± 10.5 49.2 ± 10.5	692 (60.6) 965 (61.9)	6.4 (5.4–7.6) 6.7 (5.6–7.9)	641 (56.2) 853 (54.7)	uHR 0.64 (0.45–0.93) aHR 0.66 (0.46–0.96) PHR 0.68 (0.49–0.99) C_HR 0.65 (0.45–0.94) C_aHR 0.64 (0.43–0.95)
Gordon 2019 ^†^ U.S.A. [23]	2005–2017	3.2 years (TDF + ETV)	NA	415 407	49.5 ± 11.3		NA	NA	aHR 0.73 (0.29–1.84)
Kim, 2018 Korea [24]	2007.01–2017.04	33 (21–46) 66 (36–88)	267 (44.2) 346 (48.0)	604 721	50 ± 11 50 ± 11	363 (60.1) 471 (65.3)	6.0 ± 1.6 6.4 ± 1.4	376 (62.3) 430 (59.7)	HR 1.36 (0.72–2.56) aHR 1.71 (0.79–3.70) C_HR 0.96 (0.5–1.84) C_aHR 1.47 (0.65–3.30) PHR 1.89 (0.8–4.5) C_PHR 1.25 (0.51–3.09)
Shin, 2020 Korea [25]	2007.01–2018.01	3.8 (2.7–5.0) years 6.9 (4.3–8.8) years	375 (41.67) 440 (49.22)	900 894	51 ± 11 52 ± 11	571 (63.44) 597 (66.78)	5.22 (3.32–6.97) 6.45 (5.32–7.81)	565 (62.78) 537 (60.07)	uHR 0.538 (0.352–0.822) aHR 0.752 (0.489–1.155) PHR 0.769 (0.460–1.288)
Kim 2019 Korea [26]	2012.01–2014.12	59.2 (Median) (TDF + ETV)	Compensated 411 (29.1) 499 (33.6)	1413 1484	48.8 ± 12.0 48.2 ± 11.5	913 (64.6) 889 (59.9)	5.4 ± 2.1 5.7 ± 2.1	694 (49.1) 758 (51.1)	HR 0.917 (0.705–1.191) aHR 0.975 (0.747–1.272) PHR 1.021 (0.773–1.349) IHR 0.998 (0.771–1.293) C_HR 0.848 (0.621–1.158) C_aHR 0.831 (0.606–1.139) C_aHR 0.854 (0.612–1.193) C_IHR 0.824 (0.605–1.123)
Lee 2019 Korea [27]	2007.02–2018.01	Mean 36.4 Median 36.6 Mean 60 Median 51.5	563 (39.12) 640 (40.43)	1439 1583	47.29 ± 11.16 46.66 ± 11.76	841 (58.44) 926 (58.50)	6.41 (5.34, 7.49) 6.49 (5.28, 7.67)	823 (57.19) 974 (61.53)	HR 0.912 (0.638–1.303) aHR 0.971 (0.676–1.396) PHR 1.03 (0.703–1.509) PaHR 1.077 (0.518–2.241) C_HR 0.923 (0.420–2.028) C_aHR 0.99 (0.66–1.48) C_PHR 0.956 (0.614–1.488) C_PaHR 1.077 (0.435–2.662)
Tsai 2017 Taiwan [28]	2007.01–2013.12	20.3 ± 6.4 43.8 ± 18.2	100	83 359	54.9 ± 10.9 57.8 ± 10.8	64 (77.1) 258 (71.9)	6.4 ± 1.2 6.3 ± 1.3	19 (23) 84 (23)	HR 0.52 (0.12–2.22)
Yip 2019 ^‡^ Hongkong [29]	2008.01–2018.06	2.8 years 3.7 years	35 (3) 3650 (13)	1309 28041	43.2 ± 13.1 53.4 ± 13.0	591 (45.1) 18094 (64.5)	5.3 4.8	723 (55) 8306 (30)	sHR 0.15 (0.07–0.29) asHR 0.33 (0.16–0.67) PsHR(1:1) 0.39 (0.18–0.84)
Yu 2018 Korea [30]	2007.01–2015.12	33.6 (6.3–60.5) 69.9 (6–119.4)	77 (43.8) 148 (36.5)	176 406	49 (20–84) 53 (18–84)	104 (59.1) 272 (67.0)	NA	104 (59.1) 212 (52.2)	HR 1.39 (0.56–3.45)
Yu 2019 Korea [31]	2007.02–2017.01	48.6 (29–69.7) (TDF + ETV)	371 (39.3) (TDF + ETV)	342 601	50 (41–57) (TDF + ETV)	586 (62) (TDF + ETV)	NA	528 (55.9) (TDF + ETV)	HR 1.39 (0.658–2.941)
Wu 2017 Taiwan [32]	(T)2011.10–2014.01 (E)2007.01–2012.01	37.9 ± 7.2 49 ± 19.1	29 (27.4) 94 (30)	106 313	47.1 ± 12.1 47 ± 12.3	74 (69.8) 230 (73.5)	7.35 ± 0.7 7.18 ± 0.74	50 (47.1) 172 (55)	HR 0.73 (0.26–2.05)
Hsu 2019 ^‡^ Worldwide [33]	2005.04.07–2018.12.23	38.7 (23.8–56.2) 60 (39.6–60)	131 (18.7) 1344 (27.8)	700 4837	45.74 ± 0.47 50.81 ± 0.17	456 (65.1) 3328 (68.8)	4.99 ± 0.09 5.48 ± 0.03	208 (33.7) 1537 (33.0)	sHR 0.45 (0.26–0.79) asHR 0.81 (0.42–1.56) PsHR 0.77 (0.37–1.60) PasHR 0.89 (0.41–1.92) C_sHR 0.68 (0.27–1.68)
Cho 2018 ^†^ Korea [34]	NA	NA	NA	217 517	NA	NA	NA	NA	HR 0.47(0.16–1.37)
Ha 2020 ^‡^ Korea [35]	2008.11–2017.12	NA	39 (9.3) 259 (28)	419 921	45 ± 16 48 ± 15	266 (63) 558 (61)	6.67 (2.63) 6.36 (2.31)	261 (62) 488 (53)	PsHR 2.06 (0.98–4.33) PasHR 1.84 (0.9–3.79)
Oh 2020 Korea [36]	(T)2012.01–2015.12 (E)2011.01–2014.01	Mean (years) 4.5 ± 1.1 Median (years) 4.7 (3.8, 5.4] Mean (years) 4.7 ± 1.0 Median (years) 4.9 [4.4, 5.5]	310 (38.4) 315 (41.8)	807 753	46.3 ± 11.2 48.7 ± 11.4	503 (62.3) 480 (63.7)	6.6 [5.5, 7.7] 6.5 [5.4, 7.6]	484 (60.0) 451 (61.4)	HR 1.26 (0.81–1.97)
Ha 2020 ^§^ Korea [37]	2010–2015	49.1 (37.7–62.2) 64.0 (30.5–84.3)	78 (34.8) 67 (37.2)	224 180	44.5 ± 11.4 45.4 ± 10.8	120 (53.6) 106 (58.9)	7.44 (6.33, 8.53) 7.71 (6.74, 8.64)	128 (57.1) 118 (67.4)	HR 0.31 (0.12–0.79) PHR 0.27 (0.08–0.98) IHR 0.32 (0.13–0.80) C_HR 0.30 (0.11–0.84)
Hu 2020 Taiwan [38]	2008.01–2018.03	NA (5 years sub cohort)	100%	216 678	56.1 ± 11.6 59.4 ± 11.1	162 (75) 491 (72.4)	NA	41 (19.0) 125 (18.4)	aHR 0.66 (0.40–1.08) ^‖^ PaHR 0.66 (0.38–1.14) ^‖^
Chen 2020 Taiwan [39]	(T) 2011–2018 (E) 2008–2018	NA	NA	1353 (TDF + ETV)	NA	NA	NA	NA	HR 0.523 (0.363–0.752) aHR 0.582 (0.401–0.843) C_HR 0.534 (0.355–0.805) C_aHR 0.576 (0.379–0.877)
Na 2021 * Korea [40]	2012.06–2015.12	3.8 (2.9, 4.9) (years) 5.2 (3.4, 6.2) (years)	302 (45.4) 377 (56.2)	665 671	49 (42, 56) 51 (44, 57)	384 (57.7) 392 (58.4)	5.9 (4.6, 7.1) 5.7 (4.6, 6.6)	291 (43.7) 196 (29.2)	HR 0.94 (0.63–1.41) aHR 0.82 (0.52–1.29) PHR 1.02 (0.68–1.52) IHR 1.11 (0.74–1.66)

TDF(T), tenofovir; ETV(E), entecavir; HBV, hepatitis B virus; CHB, chronic hepatitis B; M, male; F, female; NA, not available; HCC, hepatocellular carcinoma; HR; hazard ratio, uHR, univariate HR, aHR; adjusted HR, PHR; HR from propensity score matched analysis, IHR; HR from inverse probability of treatment weighting analysis, C_HR; HR from cirrhosis sub cohort, sHR; sub distribution HR. ^†^ abstract, ^‡^ suggest outcomes from competing risk analysis, ^§^ Ha from CHA Bundang Medical Center, CHA University, ^‖^ From cox regression analyses of sub-cohort of treatment-naïve patients followed up to 5 years; * from unadjusted cohort at the time of CVR.

**Table 2 cancers-14-02617-t002:** Inclusion and exclusion criteria used in the included studies.

Author Year Country	Inclusion Criteria	Exclusion Criteria
Choi 2019 Korea [7]	Treatment naïve patients Treatment > 6 months Korean adults (Age ≥ 20 and ≤ 79 years)	Serum HBV DNA at baseline <2000 IU/mL (or undetectable) More than two weeks of previous treatment with other antiviral agents Loss of hbsag within 6 months of treatment initiation Death or liver transplantation within 6 months of treatment HCC diagnosis within 1 year of treatment initiation Co-infection with human immunodeficiency virus or other hepatotrophic viruses History of any malignant disease
Gordon 2019 ^†^ U.S.A. [23]	Treatment naïve 80% (642/822)	Liver transplantation HIV co-infection ETV and TDF combine therapy
Kim, 2018 Korea [24]	Treatment naïve patients Age ≥ 18 Treatment > 12 months	Co-infection with human immunodeficiency virus or other hepatotrophic viruses Serum HBV DNA at baseline <2000 IU/mL Creatinine > 1.5 mg/dL Death within 6 months of treatment HCC diagnosis within 1 year of treatment initiation History of HCC before treatment Liver transplantation Decompensated LC patients Adherence rate < 80%
Shin, 2020 Korea [25]	Treatment naïve patients Age ≥ 18 Treatment > 12 months	Co-infection HCV, HDV, HIV History of any malignant disease Decompensated LC patients or Child–Pugh score ≥ 7 Creatinine > 1.5 mg/dL Death within 6 months of treatment HCC diagnosis within 1 year of treatment initiation
Kim 2019 Korea [26]	Treatment naïve patients Age ≥ 19 Treatment > 12 months	Coinfection with other hepatitis virus Prior organ transplant or hcc HCC, liver transplant, or death < 6 months after enrolment Decompensated cirrhosis Significant medical illness
Lee 2019 Korea [27]	Treatment naïve patients Treatment > 6 months	Co-infection with HCV, HIV HCC, liver transplant < 6 months after enrolment History of any malignant disease Prior organ transplant or HCC Decompensated cirrhosis
Tsai 2017 Taiwan [28]	Treatment naïve patients Cirrhotic patients only HBV DNA ≥ 2000 IU	HIV, HCV, HDV, HEV coinfection HCC < 6 months after enrolment History of HCC before treatment DILI/alcohol > 50 g/d
Yip 2019 Hongkong [29]	Treatment naïve patients Treatment > 6 months	HIV, HCV, HDV coinfection Autoimmune disease, metabolic liver disease History of HCC before treatment History of any malignant disease HCC, death < 6 months after enrolment Prior liver transplant or liver transplant < 6 months after enrolment ETV and TDF combined therap
Yu 2018 Korea [30]	Treatment naïve patients Age ≥ 18 Treatment > 6 months	Other viral hepatitis, Autoimmune disease, Metabolic liver disease History of HCC before treatment History of any malignant disease
Yu 2019 Korea [31]	Treatment naïve patients Age ≥ 18 Treatment > 12 months	Other viral hepatitis, autoimmune disease, metabolic liver disease History of HCC before treatment History of any malignant disease
Wu 2017 Taiwan [32]	Treatment naïve patients Treatment > 12 months High viral load >6 log10 (IU/mL)	Co-infection Alcoholic, autoimmune hepatitis HCC < 12 months of enrolment
Hsu 2019 Worldwide [33]	Treatment naïve patients Age ≥ 18 Monotherapy treatment > 12 months	Any malignant disease at the initiation HCC or death < 12 months of enrolment History of solid organ transplantation or significant use of immunosuppression Co-infection ETV and TDF combined therapy
Cho 2018 ^†^ Korea [34]	Treatment naïve patients	Na
Ha 2020 ^ǂ^ Korea [35]	Treatment naïve patients Age > 18 Treatment > 12 months	Serum HBV DNA at baseline <2000 IU/mL HCC, death, liver transplant < 6 months of enrolment Prior liver transplant or HCC before inclusion
Oh 2020 Korea [36]	Treatment naïve patients Age ≥ 18 Treatment > 12 months	Co-infection HCV, HIV History of any malignant over the preceding 5 years HCC, death, treatment modification < 12 months after enrolment
Ha 2020 ^§^ Korea [37]	Treatment naïve patients Age 18–80 years, Treatment > 6 months	Co-infection with other viral infection HCC, seroconversion, any malignancy, organ transplant < 6 months of enrolment
Hu 2020 Taiwan [38]	Cirrhotic patients only (Treatment naïve sub cohort) Treatment > 6 months	Co-infection HCV, HDV, HIV HCC < 6 months after enrolment ETV and TDF combined therapy or switching History of HCC before treatment Decompensated LC patients Alcoholic, autoimmune disease
Chen 2020 Taiwan [39]	Cirrhotic patients only Treatment naïve patients Age ≥ 18 Monotherapy treatment > 12 months	Co-infection HCV, HDV, HIV Alcoholic, autoimmune disease Liver transplant or HCC before inclusion or < 12 months after enrolment
Na 2021 Korea [40]	Treatment naïve patients Follow-up duration > 12 months Age ≥ 18	Co-infection HCV, HIV Liver transplant or HCC before inclusion or < 12 months after enrolment Prior or concurrent malignancy including HCC and organ transplantation Did not achieve MVR during NAs therapy (<20 IU/mL) Development of HCC or received liver transplantation before CVR or within one year after achieving CVR Incident malignancy other than HCC during follow-up Follow-up duration less than one year after achieving CVR Switch to other NAS

TDF, tenofovir; ETV, entecavir; CHB, chronic hepatitis B; HCV, hepatitis C virus; HDV, hepatitis D virus; HIV, human immunodeficiency virus; NA, not available; HTN, hypertension; HCC, hepatocellular carcinoma; LC, liver cirrhosis; DILI, drug induced liver injury; MVR, maintained virologic response; NAs, Nucleos(t)ide analogues; CVR, complete virologic response; ^†^ abstract, ^‡^ suggest outcomes from competing risk analysis, ^§^ Ha from CHA Bundang Medical Center, CHA University

**Table 3 cancers-14-02617-t003:** Subgroup analysis comparing clinical outcomes based on the inclusion and exclusion criteria.

Subgroup	Number of Studies	HR (95% CI)	I^2^ (%)
Treatment duration less than 6 months vs. 12 months			
6 months	6	0.56 (0.34–0.92) *	80 ^†^
12 months	10	0.83 (0.63–1.08)	66 ^†^
Exclusion of patients diagnosed with HCC within 6 months vs. 12 months			
6 months	7	0.64 (0.39–1.04)	82 ^†^
12 months	8	0.69 (0.54–0.88) ^†^	54 *
Interval of over three years in the start point of patient enrolment			
<3 years	5	0.83 (0.62–1.12)	61 *
>3 years	13	0.69 (0.51–0.92)/	68 ^†^
Exclusion of patients with baseline HBV DNA levels of <2000 IU/mL			
Yes	4	0.87 (0.50–1.52)	63
No	13	0.69 (0.52–0.90) ^†^	74 ^†^
Exclusion of patients with significant alcoholic liver disease			
Yes	4	0.58 (0.44–0.76) ^†^	0
No	11	0.75 (0.57–1.00)	76 ^†^
Exclusion of patients with CKD or baseline creatinine >1.5 mg/dL			
Yes	3	0.74 (0.51–1.05)	55
No	14	0.72 (0.53–0.97) *	74 ^†^

* *p* value < 0.05;/*p* value = 0.01; ^†^ *p* value < 0.01.

## Data Availability

All relevant data are included in the study and Appendix A.

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
