# Peer review of "Systematic Review with Meta-Analysis: Comparison of the Risk of Hepatocellular Carcinoma in Antiviral-Naive Chronic Hepatitis B Patients Treated with Entecavir versus Tenofovir: The Devil in the Detail"

_cancers, 2022, doi:10.3390/cancers14112617_

Round 1
Reviewer 1 Report
The main limitation of the article is that it covers almost exclusively Korean works. The authors did not take into account the key relationship with HBe status, and as it is known, in the years from which the works are derived, HBeAg-positive patients predominated in Asia, which is now rare, especially in rest of the world. HBe status obviously influences the course of the disease and HCC risk, so if it is omitted, the work may be of historical value at best. Authors should:
1) carry out the analysis from the beginning, taking into account the HBe status,
2) change the content of the title, abstract, introduction, methodology, results and discussion so that it clearly indicates that the work concerns Korean patients and what HBe population it covers.
Author Response
1,2) Thanks for the good point. As we mentioned in results and discussion, 12 out of 19 studies were conducted in Korea which accounted for 63%. In order to enhance understanding of the impact of Korean research, we will present the number of CHB patients of major countries in this study : Hong Kong 30,858, South Korea 18,684, Taiwan 5,565 / U.S.A. 1819. Therefore, the characteristics of patient in this study may be different from Caucasian cohort. Referring to recent research comparing Korean and Caucasian cohort (PMID: 34500103), among the 9143 patients of the entire Korean cohort (mean age, 49.2 years; male, 60.3%), 4492 (49.1%) were HBeAg-positive. As the HBeAg-positive phase occurs early in the natural course, HBeAg-positive group had younger age (47.1 years), lower prevalence of LC (41.1%), higher serum HBV DNA levels (7.2 log10 IU/mL). On the other hand, the Caucasian cohort (n = 719; mean age, 51.8 years) showed HBeAg-positive 20.3%, cirrhosis 34.8%. In above study, HBeAg positivity was independently associated with a lower risk of HCC in Korean cohort. Compared to Caucasian cohort, the proportion of LC was higher, so even if the HBeAg positivity was higher, HCC occurred more at Korean cohort. In Table 1, Except for one study dealt with only cirrhosis patients (Hu 2020), all of the studies included in this study have high HBeAg positivity. Nevertheless, the age and proportion of LC, which are impotant independant risk factors for HCC, are commonly higher than the characteristics of Caucasian cohort presented in the above study. Since we did not designed individual data meta-analysis, it is not possible to analyze which factor had a predominant effect. Although the analysis method is different, in a similar meta-analysis study, proportion of HBeAg positivity alone did not affect the efficacy of TDF and ETV on incidence of HCC in meta-regression analysis (Coefficient 0.98 (SE 0.48), 95% CI 0.39–2.50, p-value 0.968) (PMID: 32407970). Therefore, it should be understood that not only HBeAg status but also various characteristics such as age, LC, and genotype will be different. I will specify that these risk factors should be taken into account when interpreting the results in discussion.
Reviewer 2 Report
The work is very good, interesting and necessary.The novelty of this study is analysis data from 206 only antiviral-naïve CHB patients.
I recommend its publication.
Author Response
Thank you for your good comments and thank you again for your work as reviewer.
Reviewer 3 Report
The possibly different impact of the antiviral therapy with Tenofovir vs Entecavir on HCC prevention in patients with chronic hepatitis B is an open issue. The Authors conducted a new systematic review with metanalysis to further explore this question.
Some observations and comments.
The Authors claim: "The novelty of this study lies in the fact that we extracted and analysed data from only antiviral-naïve CHB patients."
However, the results of the study remain not conclusive because of the presence of other multiple variables influencing the HCC risk, that may be not adequately considered in the metanalysis.
The Authors discussed the various limits of the available published data and of the statistical methods used to evaluate the different efficacy of the two drugs. At least some minimal parameters should be consider in the metanalysis (if available in the articles included):
- The stage of liver disease conditions the risk of HCC, especially significant in cirrhotics and even more in advanced cirrhosis; in chronic hepatitis patients without a definite diagnosis of cirrhosis, the stage of fibrosis and the the risk of HCC may vary significantly. To limit this variability the metanalysis should consider only patients with compensated cirrhosis at the start of antiviral therapy.
- The presence of active cofactors (i.e. alcohol, diabetes, metabolic syndrome, ...) of liver damage may favour progression of liver disease and HCC development: this parameter is not always available
- In most studies included in the metanalysis the number of patients and the duration of the follow up in Tenofovir and Entecavir group are significantly different.
The PSM-analysis may only partially correct the bias of the retrospective studies. Therefore, the metanalysis of the studies published so far cannot lead to a definitive answer. As a consequence the Authors should more clearly declare that today, due to the limits examined in the discussion, there are not sufficient data to establish a different performance of TDV and ETV in prevention of HCC (see Abstract: "TDF might be more effective than ETV in reducing HCC incidence in treatment-naive CHB patients but not in the PS-matched subpopulation" and Conclusions: "In conclusion, our analysis found that the incidence of HCC following TDF mono-therapy was significantly lower than after ETV monotherapy.")
Finally, TDV and ETV are direct antiviral agents:
- The sentence "This systematic review and meta-analysis aim to ......... obtain new insights into the efficacies of TDF and ETV for HCC treatment" is to be corrected.
- The preventive effect on HCC risk arises from antiviral activity:
- Patients with HBV-DNA <2,000 UI/mL at baseline could be either CHB patients with transient low viremia levels (as after a flare) and patients with chronic HBV infection (not CHB) with a liver disease of different etiology --> to be considered in the selection of the population to be analysed for HCC risk
- The ability to prevent HCC should depend on antiviral activity: the analysis of the preventive effect should consider the baseline HBV-DNA and the time from baseline to serum HBV-DNA clearance.
Author Response
The Authors claim: "The novelty of this study lies in the fact that we extracted and analysed data from only antiviral-naïve CHB patients."
However, the results of the study remain not conclusive because of the presence of other multiple variables influencing the HCC risk, that may be not adequately considered in the metanalysis.
The Authors discussed the various limits of the available published data and of the statistical methods used to evaluate the different efficacy of the two drugs. At least some minimal parameters should be consider in the metanalysis (if available in the articles included):
The stage of liver disease conditions the risk of HCC, especially significant in cirrhotics and even more in advanced cirrhosis; in chronic hepatitis patients without a definite diagnosis of cirrhosis, the stage of fibrosis and the the risk of HCC may vary significantly. To limit this variability the metanalysis should consider only patients with compensated cirrhosis at the start of antiviral therapy.
The presence of active cofactors (i.e. alcohol, diabetes, metabolic syndrome, ...) of liver damage may favour progression of liver disease and HCC development: this parameter is not always available
> Thank you for the good points. Unfortunately, the number of studies that presented data necessary for analysis was insufficient. No studies suggested data of stage of fibrosis using elastography. As we described in supplement table 3, only 5 studies suggested indirect marker such as FIB-4. In table 2, only 4 studies of Taiwan dealt with alcohol intake in CHB patients.
In most studies included in the metanalysis the number of patients and the duration of the follow up in Tenofovir and Entecavir group are significantly different.
> Thank you for the good point and I strongly agree. If there is a difference in the follow period, there is a difference in the occurrence of HCC. So we designed pooled analysis using HRs, not ORs.
The PSM-analysis may only partially correct the bias of the retrospective studies. Therefore, the metanalysis of the studies published so far cannot lead to a definitive answer. As a consequence the Authors should more clearly declare that today, due to the limits examined in the discussion, there are not sufficient data to establish a different performance of TDV and ETV in prevention of HCC (see Abstract: "TDF might be more effective than ETV in reducing HCC incidence in treatment-naive CHB patients but not in the PS-matched subpopulation" and Conclusions: "In conclusion, our analysis found that the incidence of HCC following TDF mono-therapy was significantly lower than after ETV monotherapy.")
> Thank you for the good point. I will specify that it is an pooled analysis of HR before reducing heterogeneity. I agree with the limitations of retrospective cohort study.
Finally, TDV and ETV are direct antiviral agents:
The sentence "This systematic review and meta-analysis aim to ......... obtain new insights into the efficacies of TDF and ETV for HCC treatment" is to be corrected.
> Thank you for the good point. We corrected it.
The preventive effect on HCC risk arises from antiviral activity:
Patients with HBV-DNA <2,000 UI/mL at baseline could be either CHB patients with transient low viremia levels (as after a flare) and patients with chronic HBV infection (not CHB) with a liver disease of different etiology --> to be considered in the selection of the population to be analysed for HCC risk.
> Thank you for the good point. Low level viremia is one of the most important issues in recent years. In table 2, some of the studies excluded CHB patients with HBV-DNA < 2,000 UI/mL. There may be differences in the outcomes as we mentioned in table 3. However, in Korea, when excluding HBV-DNA < 2,000 UI/mL, all decompensated cirrhosis patients would be excluded from the cohort after 2015.09.01 according to the reimbursement policy (supplement table 7). Since we did not designed individual data meta-analysis, the effect of the HBV-DNA level criteria could not be analyzed. We discussed this issue in discussion, row 268-272.
The ability to prevent HCC should depend on antiviral activity: the analysis of the preventive effect should consider the baseline HBV-DNA and the time from baseline to serum HBV-DNA clearance.
> Thank you for the good points. Unfortunately, the number of studies that presented data necessary for analysis was insufficient. 7 studies suggested data of virologic response, but only one study presented time to CVR (Na 2021).
Reviewer 4 Report
Hyunwoo Oh et al. uncovered a comparison of the risk of hepatocellular carcinoma in antiviral-naive chronic hepatitis B patients treated with entecavir versus tenofovir.
Points to be addressed:
1) The rationale of why the authors came up with this review.
2) What is the information that is not exactly available that motivated the authors to come up with this information. What are the current caveats and how do the authors highlight the current research in answering them? If not they need to address in future directions.
3)This reviewer personally misses some insights regarding the novel role of endothelial cells view as gatekeeper for HCC immunity: as is now well known, tumors grow and evolve through a constant crosstalk with the surrounding microenvironment, and emerging evidence indicates that angiogenesis and immunosuppression frequently occur simultaneously in response to this crosstalk. Accordingly, strategies combining anti-angiogenic therapy and immunotherapy seem to have the potential to tip the balance of the tumor microenvironment and improve treatment response.
4) In the frame of point 3 thinking, the few therapeutic strategies for advance hepatocellular carcinoma (HCC) on poor knowledge of its biology pointed toward, for several years, sorafenib, as na tyrosine kinase inhibitors (TKI) inhibitor,whose activity is the inhibition of the retrovirus-associated DNA sequences protein (RAS)/Rapidly Accelerated Fibrosarcoma protein (RAF)/mitogen-activated and extracellular-signal regulated kinase (MEK)/extracellular-signal regulated kinases (ERK) signaling pathway. However, the efficacy of sorafenib is limited by the development of drug resistance, and the major neuronal isoform of RAF, BRAF and MEK pathways play a critical and central role in HCC escape from TKIs activity. Advanced HCC patients with a BRAF mutation display a multifocal and/or more aggressive behavior with resistance to TKI. How the authors standpoint would envision an overcoming of these limitations (please refer to PMID: 31766556 and expand)?
5.The great expression of immune checkpoint molecules, such as programmed death-1 (PD-1), cytotoxic T-lymphocyte antigen 4 (CTLA-4), lymphocyte activating gene 3 protein (LAG-3), and mucin domain molecule 3 (TIM-3), on tumor and immune cells and the high levels of immunosuppressive cytokines induce T cell inhibition and represent one of the major mechanisms of HCC immune escape. Recently, immunotherapy based on the use of immune checkpoint inhibitors (ICIs), as single agents or in combination with kinase inhibitors, anti-angiogenic drugs, chemotherapeutic agents, and locoregional therapies, offers great promise in the treatment of HCC: how would the authors findings fit with these eviodences? Please expand.
6. In the frame of point 5. thinking we have now an evolving role: hepatocellular carcinoma, immune checkpoint inhibitors, immune checkpoint molecules, immune microenvironment: these keywords have more interactions among themselves than what would be expected few years ago. Such an enrichment indicates that are at least partially biologically connected, as a group: can the authors comment on this?
7. given the considerable complexity of the therapeutic landscape, the aim of this meta-analysis seems also consistent to compare the efficacy and safety of first but also second‐line agents backbone to drive correct interpretations and to highlight the strengths and weaknesses of the available clinical data. Moreover, this study points toward a personalized approach based on novel criteria for the management of HCC: could the authors mention and discuss similar approaches available in. this regard?
Author Response
1-2) Thanks for the good point. As I mentioned in introduction, it has yet conclusive which anti-viral agent is more effective in reducing the risk of HCC. Many previously published studies that dealt with this subject do not consider unobservable confounding factors such as reimbursement policies that can affect the heterogeneity of the studies, so we tried to present a new perception.
We targeted only anti-viral treatment naive patients to reduce heterogeneity. And we suggested subgroup analysis reflecting changes of treatment guideline and patient enrollment timing.
3-7) Thank you for the good comment that can broaden perception of HCC treatment. Unfortunately, in this study, we compared whether there is a difference in the effect of reducing HCC incidence according to anti-viral agent selection in chronic hepatitis B patients. We did not analyze factors that could affect HCC treatment response. Article (PMID: 31766556) also seems difficult to adopt as a reference in our study.
Round 2
Reviewer 1 Report
Authors did not corrected data, but provided sufficient explanation, so the article can be published in the current form.
Author Response

(The authors gave the same response as above.)

Reviewer 4 Report
The authors have clarified several of the questions I raised in my previous review. Most of the major problems have been addressed by this revision.
This reviewer still has a curiosity: commonly, HCC development occurs in a liver that is severely compromised by chronic injury or inflammation, associated with chronic infection with hepatitis B virus (HBV) or hepatitis C virus (HCV), the consumption of aflatoxins, frequent contaminants of foods and feeds that can act synergistically with HBV, and tobacco smocking. Hepatocarcinogenesis is a multistep process starting with liver injury, followed by gene mutations, chronic local inflammation, fibrosis, cirrhosis, and cancer. The introduction of anti-HBV vaccination and direct-acting viral (DAA) therapies for HCV treatment has led to the almost total eradication of the viruses, in terms of undetected viral nucleic acids. However, although HBV or HCV eradication is associated with the regression of fibrosis or cirrhosis, these patients might have a higher risk of developing HCC compared with pre-cirrhotic patients (PMID: 34065489). Can the authors spend few lines on this?
Author Response
Thank you for the good comment. First of all, patients coinfected with HCV are excluded from our research, so I'll put it aside. And very interestingly, as reviewer pointed out, despite anti-viral treatment, the risk of HCC is steadily high. In Korea, CHB infection by mainly vertical transmission (abour 95%) cause a life-long inflammation, and most of them are infected with HBV genotype C which is higher risk of hepatocellular carcinoma than other major hepatitis B virus genotypes. Thus even if patients take anti-viral drugs steadily, the risk of developing HCC is high. In addition, the increased survival period of patients due to the use of anti-viral drugs also affected the results. We briefly mentions this in introduction with reference [2, 3]. The review presented by the reviewer (PMID: 34065489) suggested a good perception related to this issue, so we take it as reference [59] in the discussion.